# Segmentation Helps Understanding: Mask-Infused Vision-Language Pre-training for 3D Medical Images

## Abstract

Pretraining effective 3D medical image encoders is essential for advancing downstream tasks such as diagnosis and prognosis. Compared to 2D images, 3D scans are extremely high-dimensional and information-sparse, which makes it especially difficult for vision–language pretraining methods to align global semantic descriptions with the fine-grained voxel patterns that underlie clinically significant findings. To address this challenge, we propose `SegVL`, a unified contrastive learning framework that incorporates segmentation data into vision–language pretraining. At its core, `SegVL` introduces voxel–mask contrastive learning, where segmentation mask names act as text anchors to guide voxel embeddings with anatomical semantics, and it leverages the same cues to enrich global image representations for stronger image–text alignment. To mitigate extreme class imbalance we adopt a Tversky loss, and to preserve encoder capacity we emphasize a lightweight decoder. Extensive experiments across multiple external datasets and tasks demonstrate that `SegVL` consistently outperforms existing pretraining strategies, validating the synergy of segmentation and vision–language data in learning comprehensive 3D medical representations.

## 1 Introduction

Pretraining powerful 3D medical image encoders is a critical challenge with the potential to significantly enhance the performance of various downstream tasks, such as diagnosis and prognosis Hamamci et al. (2024). Among the widely adopted strategies, Contrastive Language-Image Pretraining (CLIP)-style approaches, as shown in Figure 1(a), have shown promise Hamamci et al. (2024); Blankemeier et al. (2024); Shui et al. (2025). These methods leverage the information present in paired radiology reports to supervise the visual encoder, enabling it to capture semantic information within images. While these pretrained models have demonstrated improved performance on downstream classification tasks, they often primarily focus on high-level semantic features and tend to overlook the clinically critical fine-grained information inherent in high-dimensional 3D medical images. For instance, a typical thin-slice CT scan contains over 52 million voxels (512×512×200), but a high-risk lung nodule may occupy less than 0.005% of the entire volume. This significant disparity in scale makes it exceptionally challenging for existing Vision-Language Pretraining (VLP) models to effectively capture such subtle yet clinically vital details.

Several works have attempted to address the above issue, particularly by introducing multi-granularity alignment during pretraining Wang et al. (2022a); Boecking et al. (2022); Cheng et al. (2023). These methods primarily focus on algorithm modifications rather than incorporating additional data. Consequently, they are predominantly applied to 2D data, as extending them to 3D medical images, which exhibit an order of magnitude higher dimensionality and sparser information, often proves insufficient for capturing correct detailed information without external data. While targeted annotations containing specific locations described in reports are scarce and challenging to acquire, we posit that fine-grained features may inherently exist within readily available segmentation data that is not directly linked to the reports. For instance, nodules are delineated in segmentation masks, and segmentation is frequently considered a crucial pre-processing step in various real-world medical tasks Khan et al. (2021); Zhou et al. (2019). This underexplored perspective motivates our approach to leverage segmentation data as a potential source of fine-grained visual information for pretraining.

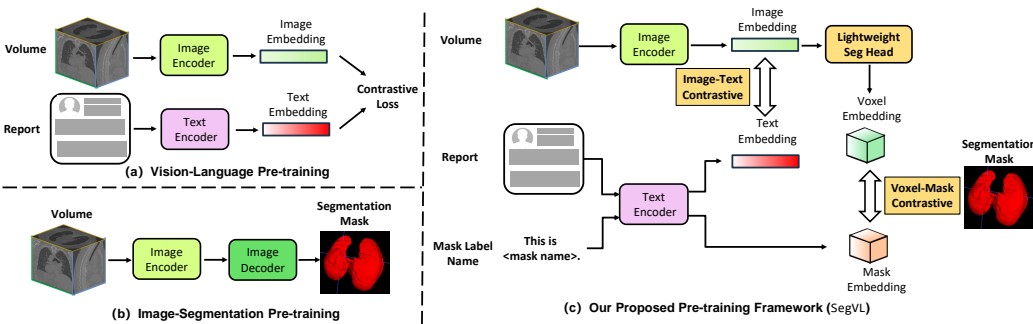

Figure 1: Overview of 3D medical image pretraining paradigms. (a) CLIP-style methods align entire volumes with reports via image-text contrastive learning, but ignore spatial granularity. (b) Image-segmentation pretraining supervises voxel-level predictions but typically uses heavy decoders. (c) Our SegVL unifies both modalities via two contrastive objectives: image-text and voxel-mask. A lightweight segmentation head produces voxel embeddings, and mask embeddings are obtained from shared text encoders. Segmentation masks are used for defining positive/negative voxel-mask pairs.

In this paper, we aim to explore an underexamined but valuable objective: *how to jointly leverage segmentation and VL data to build a more powerful 3D medical image encoder*. Our goal is to imbue the encoder not only with the high-level semantic understanding capabilities derived from VL data but also with the fine-grained information extraction abilities inherent in segmentation data. Enabling a single encoder to effectively benefit from both segmentation and VL data presents non-trivial challenges. Firstly, the granularity of information provided by language (high-level semantics) differs significantly from that of segmentation masks (low-level pixel details). Secondly, VLP frameworks typically employ a two-encoder architecture with a VL contrastive learning objective, whereas segmentation-based approaches commonly adopt an encoder-decoder structure trained for accurate mask prediction (shown in Figure 1(b)).

We propose a novel unified contrastive learning framework, referred to as SegVL and shown in Figure 1(c), that bridges disparate segmentation and VL data, enabling them to collaboratively contribute to a single image encoder. While retaining the principles of VL contrastive learning, we innovatively infuse segmentation data through an additional contrastive learning mechanism. Specifically, we encode segmentation mask names using the existing text encoder from the VL branch and perform *voxel-mask contrastive learning* with the voxel embeddings produced by the image encoder. To address the extreme class imbalance in segmentation data, we employ a learning objective based on the Tversky loss Salehi et al. (2017). Furthermore, we leverage the segmentation information to enhance the image representation, and subsequently use this augmented representation for *image-text contrastive learning*. This unified contrastive learning framework allows the model to effectively assimilate information from both kinds of data. Notably, we find that employing a lightweight decoder is crucial; the heavy decoders commonly used in segmentation tasks tend to absorb the relevant information, preventing it from being effectively retained within the encoder, which contradicts our goal of learning a robust and informative encoder.

In summary, our contributions are threefold: (1) We pioneer the exploration of a significant yet underexplored direction: jointly leveraging unlinked VL and segmentation data to build a powerful 3D medical image encoder with fine-grained understanding, while also improving segmentation transferability within the VLP family. (2) We propose SegVL, a novel unified contrastive learning framework that employs voxel-mask contrastive learning to effectively learn from segmentation data and enhanced image–text contrastive learning to learn from VL data. (3) We validate the effectiveness of SegVL across a broad range of downstream tasks and datasets. Our SegVL consistently improves understanding-oriented tasks: fine-tuning on external datasets yields an average AUC gain of 1.8% over previous SOTA across multiple classification benchmarks. At the same time, SegVL also enhances segmentation transferability (e.g., +3.2% Dice in lung tumor segmentation) of VLP, showing that the learned representations are both contextually rich and spatially precise.

## 2 RELATED WORK

**Medical Vision-Language Pre-training** Medical vision-language pretraining (Med-VLP) aims to learn generalizable image representations by aligning medical images with paired radiology reports. Early works in this area have primarily focused on 2D imaging modalities, such as chest X-rays, leveraging contrastive learning between images and free-text reports Zhang et al. (2022; 2023); Boecking et al. (2022); Wang et al. (2022b). Improvements in this direction include the use of domain-specific pretraining of text encoders Boecking et al. (2022), knowledge-enhanced contrastive objectives Wu et al. (2023), and hierarchical attention mechanisms to better align visual and textual information Huang et al. (2021); Wang et al. (2022a); Cheng et al. (2023). These methods have demonstrated strong performance in classification, retrieval, and report generation tasks. However, most of these works are limited to 2D images with relatively low resolution and compact spatial content. In contrast, 3D medical images, such as CT volumes, present significantly greater challenges due to their higher dimensionality and sparser distribution of clinically relevant information.

Recent Med-VLP efforts have begun extending to 3D modalities, leveraging the greater anatomical detail in volumetric data. Merlin Blankemeier et al. (2024) explores abdominal CT scans paired with electronic health records. CT-CLIP Hamamci et al. (2024) was the first to introduce a public large-scale benchmark dataset of chest CT volumes paired with radiology reports, along with a 3D vision-language pretraining model. To address misalignment problem caused by global image-text contrastive learning Zhang et al. (2022), several recent methods have explored more detailed supervision. On the image side, T3D Liu et al. (2023) aligns sub-volumes across different views of the same scan, and CT-FM Pai et al. (2025) introduces 3D-patch-level contrastive learning. On the text side, MG-3D Ni et al. (2024) models the alignment between CT and each sentence within the reports. fVLM Shui et al. (2025) further combines both sides by extracting organ-specific sub-volumes via segmentation and aligning them with organ-level text descriptions parsed by an LLM. Unlike previous work, our method directly leverages segmentation data for supervision during pretraining, which simplifies the pipeline and enables finer-level alignment. While prior methods focus on the sub-volume level, our approach targets alignment at the voxel level.

**Utilizing Segmentation Data in Medical Image Pre-Training** Segmentation annotations are valuable for medical image analysis, facilitating the localization and understanding of organs, lesions, and other anatomical structures. As segmentation requires voxel-level predictions, segmentation data naturally contain fine-grained features critical for detailed image understanding. MedicalNet Chen et al. (2019) pretrains 3D encoders using multi-organ segmentation datasets to improve downstream performance. More common usage for segmentation data is pre-training models dedicated to segmentation tasks, like SAM Kirillov et al. (2023) and its medical adaptations Zhu et al. (2024); Wang et al. (2023). Nonetheless, our work focuses on infusing segmentation data within a VL framework, building upon the established foundation of VLP for encoder pretraining. Furthermore, while existing segmentation models typically rely on heavy decoders to reconstruct dense predictions, we have identified the significant role of lightweight decoders in the pretraining of 3D medical image encoders.

## 3 METHODOLOGY

Overall, `SegVL` is a kind of VLP framework, whose innovation lies in infusing segmentation data into the pretraining process through a unified contrastive learning approach. In this section, we first detail the encoding of 3D medical images and their corresponding textual reports, which forms the basis of our VLP framework. Subsequently, in the second subsection, we elaborate on how we effectively incorporate segmentation data by adapting the existing text encoder and employing a contrastive learning strategy. Finally, the third subsection describes our VL pretraining component, which is further augmented by the inclusion of segmentation information to facilitate a more comprehensive understanding of the visual and textual modalities.

### 3.1 IMAGE AND REPORT FEATURE EXTRACTION

Given a paired 3D medical volume $V_i \in \mathbb{R}^{H \times W \times D}$ and its corresponding radiology report $R_i$ where $i$ is the sample pair index, we first extract their representations using modality-specific encoders.

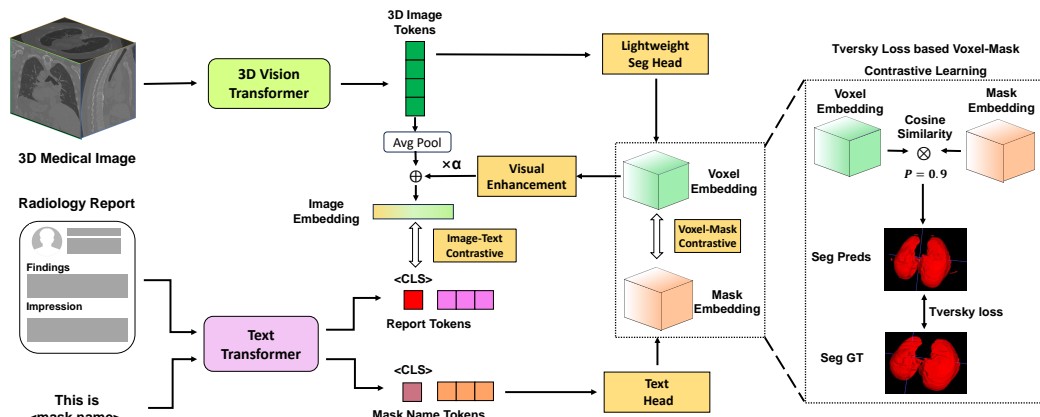

Figure 2: Architecture of our `SegVL` framework. The model jointly leverages image-report and image-segmentation data via two contrastive objectives. A 3D vision transformer encodes volumes into tokens and a lightweight segmentation head converts tokens into voxel embeddings. Voxel embeddings are supervised by voxel-mask contrastive learning using Tversky loss to handle class imbalance. Meanwhile, segmentation features are fused into global image embeddings through a visual enhancement module, which are contrasted with report embeddings to learn both high-level semantics and fine-grained anatomical cues.

**3D Image Encoding**    We use a 3D Vision Transformer $f_v(\cdot)$ to encode the volume into a set of spatial tokens:

$$\mathbf{X}_i = f_v(V_i) \in \mathbb{R}^{h \times w \times d \times d_f}, \tag{1}$$

where $h, w, d$ are the downsampled spatial dimensions, and $d_f$ is the token feature dimension. We then apply global average pooling and a following lightweight MLP projection $p_{\text{img}}$ to obtain a volume-level representation:

$$\mathbf{z}_i^{\text{img}} = p_{\text{img}}(\text{AvgPool}(\mathbf{X}_i)) \in \mathbb{R}^{d_f}.$$

**Report Encoding**    For text, we initialize a transformer-based encoder $f_t(\cdot)$ from BioViL Boecking et al. (2022), and apply it to the report $R_i$ to obtain token embeddings:

$$\mathbf{T}_i = f_t(R_i) \in \mathbb{R}^{L_r \times d_r}, \tag{2}$$

where $L_r$ is the number of tokens and $d_r$ is the token embedding dimension. We extract the [CLS] token and project it to the shared space via another MLP $p_{\text{rep}}$:

$$\mathbf{z}_i^{\text{rep}} = p_{\text{rep}}\left(T_i^{[\text{CLS}]}\right) \in \mathbb{R}^{d_f}, \quad \text{where} \quad T_i^{[\text{CLS}]} \in \mathbb{R}^{d_r}.$$

## 3.2 Voxel-Mask Contrastive Learning for Infusing Segmentation Data

To enable fine-grained voxel-level supervision, we design a contrastive learning mechanism that leverages segmentation masks. Specifically, we map transformer tokens to per-voxel embeddings to enable learning from voxel-level masks. Instead of predicting segmentation masks with decoders, we leverage the existing text encoder and employ a voxel-mask contrastive learning approach to enable the encoder learning from segmentation data with different mask names. This approach helps refine the voxel-level semantic understanding by aligning the voxel embeddings with the segmentation label embeddings. We first illustrate how voxel and mask name embeddings are obtained, and then explain the mechanism of voxel-mask contrastive learning and its rationales.

**Voxel Embeddings**    Given the 3D image tokens extracted from the vision encoder (i.e., $\mathbf{X}_i$ in Eq. 1), we apply a *lightweight segmentation head* to convert tokens to voxel embedding tensor. Here, the segmentation head is a shared MLP $f_{\text{seg}}(\cdot)$ for every tokens and a reshaping operation.

$$\hat{\mathbf{X}}_i = f_{\text{seg}}(\mathbf{X}_i) \in \mathbb{R}^{h \times w \times d \times d_{\text{mlp}}}$$
$$\mathbf{V}_i = \text{reshape}(\hat{\mathbf{X}}_i) \in \mathbb{R}^{H \times W \times D \times n_{\text{logits}}}, \tag{3}$$

where the original voxel grid size is $H \times W \times D$ and each voxel in the 3D volume has a corresponding embedding of size $\mathbb{R}^{n_{\text{logits}}}$. It is worth noting that here we opt for a lightweight MLP as the segmentation head, instead of using multi-layer decoders with a similar capacity to the encoder, as done in common works on segmentation mask prediction Li et al. (2023); Hatamizadeh et al. (2022). This design choice effectively prevents the information in the data from being absorbed by the decoder rather than being learned by the encoder, which well aligns with our objective of learning a good encoder (see experiments in Table 6).

**Mask Name Embeddings**    To obtain the mask name embeddings, we use the textual descriptions corresponding to each class in the segmentation task. For each of the $C$ segmentation classes, the mask name is described using the prompt $\mathbf{p}_c =$ "This is <mask name>", where '<mask name>' refers to the anatomical label of the class. These prompts are passed through the same text transformer in report encoding (i.e., $f_t(\cdot)$ in Eq. 2), to obtain the mask name embeddings:

$$\mathbf{M} = f_t(\mathbf{p}_c) \in \mathbb{R}^{C \times L \times d_f},$$

where $L$ is the number of tokens in the description and $d_f$ is the embedding dimension. To obtain the final class-specific mask embeddings, we extract the [CLS] token for each class (which corresponds to the embedding of size $C \times d_f$) and pass it through an text head $f_{\text{text}}(\cdot)$, which projects the embeddings into a shared embedding space of size $n_{\text{logits}}$:

$$\mathbf{M}^{\text{mask}} = f_{\text{text}}(\mathbf{M}) \in \mathbb{R}^{C \times n_{\text{logits}}},$$

where $C$ represents the number of classes and $n_{\text{logits}}$ is the size of the final mask embedding.

**Voxel-Mask Contrastive Learning**    We perform contrastive learning between voxel embeddings and mask label names. A straightforward approach is to use InfoNCE loss Oord et al. (2018) to align voxel and mask name embeddings. However, since voxels labeled as foreground (i.e., belonging to the class indicated by the mask name) form positive pairs, while background voxels serve as negatives, we observed that InfoNCE loss performs poorly in this context due to the extreme class imbalance inherent in segmentation masks (see Section 6 for experiment results). Therefore, we employ a modified contrastive learning objective based on the Tversky loss. Specifically, we first calculate the cosine similarity between the voxel embedding $\mathbf{V}_i \in \mathbb{R}^{n_{\text{logits}}}$ and the mask embeddings $\mathbf{M}_j^{\text{mask}} \in \mathbb{R}^{n_{\text{logits}}}$ as:

$$P_{i,j} = \frac{1}{2}\left(\frac{\mathbf{V}_i \cdot \mathbf{M}_j^{\text{mask}\top}}{\|\mathbf{V}_i\|\|\mathbf{M}_j^{\text{mask}}\|} + 1\right) \in \mathbb{R}^C, \tag{4}$$

where $\mathbf{M}_j^{\text{mask}}$ is the embedding of $j$-th mask class name. $i$ and $P_{i,j}$ represent the voxel index and the predicted probabilities in the range of [0, 1], respectively.

Then, we calculate the Voxel-Mask contrastive learning loss (denoted as $\mathcal{L}_{\text{VM}}$) based on similarity $P_{i,j}$ by utilizing the Tversky loss Salehi et al. (2017). This can handle extreme class imbalance issue, rather than treating each class uniformly in common contrastive learning frameworks.

$$\mathcal{L}_{\text{VM}} = \frac{1}{C}\sum_{j=1}^{C}\frac{\sum_{i=1}^{N}\left(\alpha\hat{y}_{i,j}(1-P_{i,j}) + \beta(1-\hat{y}_{i,j})P_{i,j}\right)}{\sum_{i=1}^{N}\left(\hat{y}_{i,j}(1-P_{i,j}) + (1-\hat{y}_{i,j})P_{i,j} + \hat{y}_{i,j}P_{i,j}\right)}, \tag{5}$$

where $\alpha$ and $\beta$ control the balance between false positives and false negatives, $\hat{y}_{i,j}$ is the true one-hot label for voxel $i$ and class $j$, $P_{i,j}$ is the predicted probability for voxel $i$ and class $j$, $C$ is the number of classes, and $N = H * W * D$ is the total number of voxels.

### 3.3 Image-Text Contrastive Learning with Visual Enhancement from Segmentation

We now detail the image-report contrastive learning component within SegVL, with a particular emphasis on how we leverage segmentation information to further enhance the image features. Specifically, tokens learned by the segmentation head (i.e., $\hat{\mathbf{X}}_i$ in Eq. 3) are passed through an MLP ($f_{\text{VE}}$) and then aggregated using average pooling to obtain a single feature vector:

$$\mathbf{z}_i^{\text{seg}} = \text{AvgPool}(f_{\text{VE}}(\hat{\mathbf{X}}_i)) \in \mathbb{R}^{d_f}.$$

The resulting pooled feature vector $\mathbf{z}_{\text{seg}}$ is combined with the original image embedding $\mathbf{z}_i^{\text{img}}$ as:

$$\mathbf{z}_i^{\text{img,seg}} = \mathbf{z}_i^{\text{img}} + \lambda \mathbf{z}_i^{\text{seg}},$$

where $\lambda$ is a learnable weight initialized with 0 that controls the contribution of the segmentation features to the final enhanced image embedding. This weighted sum enhances the image embedding, allowing it to better capture both the global structure and local details, improving the image-text alignment in the contrastive learning framework.

The overall loss function is a weighted combination of Image-Text (denoted as $\mathcal{L}_{\text{IT}}$) and Voxel-Mask ($\mathcal{L}_{\text{VM}}$ in Eq. 5) contrastive learning loss:

$$\mathcal{L}_{\text{SegVL}} = \mathcal{L}_{\text{IT}}(\mathbf{z}_i^{\text{img,seg}}, \mathbf{z}_i^{\text{rep}}) + \alpha_{\text{seg}} \mathcal{L}_{\text{VM}}.$$

Here, we use InfoNCE loss Oord et al. (2018) to implement $\mathcal{L}_{\text{IT}}$ and $\alpha_{\text{seg}}$ is a hyperparameter for balance.

## 4 EXPERIMENT

### 4.1 EXPERIMENTAL SETUP

**Pre-training Dataset** We conduct pretraining on the CT-RATE Hamamci et al. (2024) and RadGenome-ChestCT Zhang et al. (2024) datasets. CT-RATE contains a total of 50,188 reconstructed chest CT volumes from 21,304 patients, each paired with its corresponding radiology report, making it a large-scale 3D image-text dataset. RadGenome-ChestCT builds upon CT-RATE by further processing the volumes and adding diverse segmentation masks, including annotations covering 197 anatomical categories frequently referenced in clinical reports. In our experiments, we utilize segmentation masks for six major anatomical categories as our image-segmentation data source. More details are included in the Appendix D.1.

**Downstream Tasks and Datasets** We evaluate our model on three kinds of tasks. *(1) Classification:* We use four chest CT datasets with diagnostic labels. CT-RATE internal validation set Hamamci et al. (2024) and Rad-ChestCT Draelos et al. (2021) contain multi-label annotations for common thoracic abnormalities. CC-CCII Zhang et al. (2020) and RICORD Tsai et al. (2020) are designed for COVID-19 diagnosis. *(2) Report Generation:* We further evaluate whether the learned encoder representations transfer to clinically relevant report generation. Following CT-CHAT Hamamci et al. (2024), we benchmark on the CT-RATE validation set Hamamci et al. (2024), focusing on the report generation subset. To isolate the contribution of the encoder, the frozen encoder output is projected into a sequence of virtual image tokens and concatenated with instruction tokens before being fed into a pretrained frozen LLM. *(3) Segmentation:* We assess segmentation performance across two datasets. TotalSegmentator Wasserthal et al. (2023) provides whole-body organ masks, and we extract the lung region to focus on thoracic anatomy. MSD Antonelli et al. (2022) focuses on lung tumor segmentation and serves as a benchmark for fine-grained lesion detection. More results on AMOS Ji et al. (2022) dataset for domain generalization are included in Appendix G.

**Pre-processing and Implementation Details** Data pre-processing follows the CT-RATE Hamamci et al. (2024) protocol. Each CT volume is resampled to a uniform spacing of 0.75 mm in the x- and y-axes and 1.5 mm in the z-axis. Volumes are then padded to $480 \times 480 \times 240$, clipped to intensity range of $[-1000, 1000]$, and normalized to $[-1, 1]$. Other details are available in Appendix C.

**Compared Baselines** We compare SegVL with a broad range of state-of-the-art Med-VLP models spanning different image dimensionalities and alignment strategies.

**(1) 2D Med-VLP:** *MRM* Zhou et al. (2023) and *IMITATE* Liu et al. (2024) are 2D Med-VLP methods that operate primarily on chest X-rays. IMITATE introduces multi-level supervision to improve alignment granularity.

**(2) 2D-3D Unified Med-VLP:** *UniMiSS* Xie et al. (2022) aligns 2D and 3D data in a shared vision-language space by selecting text-relevant 2D slices from 3D volumes, enabling unified representation across modalities.

Table 1: Comparison of our SegVL with other baselines on downstream classification tasks under **linear probing** setting.

| Task | Rad-ChestCT | | | CC-CCII | | | RICORD | | | Mean (AUC) |
|---|---|---|---|---|---|---|---|---|---|---|
| | Prec | AUC | F1 | Prec | AUC | F1 | Prec | AUC | F1 | |
| CT-CLIP Hamamci et al. (2024) | 0.342 | 0.653 | 0.667 | 0.823 | 0.865 | 0.847 | 0.824 | 0.846 | 0.843 | 0.785 |
| Merlin Blankemeier et al. (2024) | 0.366 | 0.677 | 0.676 | 0.854 | 0.877 | 0.871 | 0.836 | 0.854 | 0.862 | 0.803 |
| UniMiSS Xie et al. (2022) | \ | \ | \ | 0.838 | 0.841 | 0.864 | 0.847 | 0.862 | 0.885 | \ |
| fVLM Shui et al. (2025) | 0.394 | 0.697 | 0.713 | 0.862 | 0.871 | 0.869 | 0.862 | 0.858 | 0.847 | 0.809 |
| SegVL (ours) | **0.431** | **0.702** | **0.725** | **0.889** | **0.893** | **0.904** | **0.884** | **0.887** | **0.895** | **0.827** |

Table 2: Comparison of our SegVL with other baselines on downstream classification tasks under **finetuning** setting.

| Task | Rad-ChestCT | | | CC-CCII | | | RICORD | | | Mean (AUC) |
|---|---|---|---|---|---|---|---|---|---|---|
| | Prec | AUC | F1 | Prec | AUC | F1 | Prec | AUC | F1 | |
| CT-CLIP Hamamci et al. (2024) | 0.346 | 0.650 | 0.677 | 0.908 | 0.920 | 0.903 | 0.858 | 0.863 | 0.840 | 0.811 |
| Merlin Blankemeier et al. (2024) | 0.381 | 0.694 | 0.688 | 0.915 | 0.919 | 0.893 | 0.866 | 0.879 | 0.850 | 0.831 |
| T3D Liu et al. (2023) | \ | \ | \ | **0.931** | 0.927 | 0.921 | \ | \ | \ | \ |
| UniMiSS Xie et al. (2022) | \ | \ | \ | 0.920 | 0.913 | 0.905 | 0.903 | 0.891 | 0.888 | \ |
| MRM Zhou et al. (2023) | \ | \ | \ | 0.852 | 0.880 | 0.885 | \ | \ | \ | \ |
| IMITATE Liu et al. (2024) | \ | \ | \ | 0.864 | 0.892 | 0.897 | \ | \ | \ | \ |
| fVLM Shui et al. (2025) | 0.417 | 0.704 | 0.704 | 0.916 | 0.926 | 0.913 | 0.875 | 0.885 | 0.869 | 0.838 |
| SegVL (ours) | **0.443** | **0.716** | **0.717** | **0.931** | **0.936** | **0.939** | **0.915** | **0.912** | **0.924** | **0.855** |

**(3) 3D Med-VLP:** *CT-CLIP* Hamamci et al. (2024) and *Merlin* Blankemeier et al. (2024) adopt CLIP-style dual encoders for global alignment between whole CT volumes and paired reports (or EHR data). *T3D* Liu et al. (2023) introduces a Text-informed Multi-view Alignment strategy to align sub-volumes from different spatial views. *fVLM* Shui et al. (2025) further improves alignment granularity by associating organ-level segmentation regions with per-organ textual descriptions extracted via large language models.

## 4.2 RESULTS ON CLASSIFICATION TASKS

We evaluate the transferability of our pretrained encoder on three external classification datasets: Rad-ChestCT, CC-CCII, and RICORD. First we evaluate the models under linear probing settings in Table 1. For multi-disease diagnosis in Rad-ChestCT, our SegVL achieves notable gains over CT-CLIP with +8.9% precision, +4.9% AUC, and +5.8% F1 score. Our method also surpasses the SOTA baseline fVLM in AUC and F1 score, showing its effectiveness in downstream diagnosis tasks.

CC-CCII and RICORD are both COVID-19 diagnosis datasets, where accurate classification depends heavily on subtle and localized imaging patterns. On CC-CCII, our method improves AUC by +2.8% over CT-CLIP and outperforms fVLM by +2.2%. On RICORD, SegVL achieves +4.1% gain in AUC and +5.2% in F1 score over CT-CLIP, while surpassing fVLM by +2.9% and +4.8%, respectively. These results highlight that our segmentation-guided pretraining enables learning fine-grained and transferable features that perform strongly with simple linear classifiers.

Further, in Table 2, we observe that finetuning unlocks greater capacity in our encoder, especially for more complex tasks. On Rad-ChestCT, SegVL brings +6.6% AUC and +4.0% F1 improvements over CT-CLIP and outperforms fVLM by +1.2% AUC. On RICORD, the benefits are more pronounced, with our model achieving +2.7% and +5.5% over fVLM. These results indicate that fine-grained supervision introduced during pretraining continues to yield significant advantages when models are further adapted to downstream targets.

On the internal CT-RATE validation set, SegVL achieves competitive performance across all evaluation settings. Compared to CT-CLIP, it improves AUC by +4.4% and +5.5% under linear probing and finetuning, respectively. While fVLM achieves the best zero-shot results, it leverages LLM-derived organ-level reports and performs per-organ inferences. In contrast, SegVL does not rely on such external supervision but still surpasses fVLM under linear probing and finetuning settings, demonstrating stronger generalization and more practical utility.

Table 3: Classification performance comparison on internal CT-RATE validation dataset.

| Dataset | CT-RATE | | | | | | | | |
|---|---|---|---|---|---|---|---|---|---|
| Task | Zero-shot | | | Linear-Probing | | | Finetuning | | |
| Metrics | Prec | AUC | F1 | Prec | AUC | F1 | Prec | AUC | F1 |
| CT-CLIP Hamamci et al. (2024) | 0.306 | 0.704 | 0.691 | 0.331 | 0.749 | 0.714 | 0.342 | 0.756 | 0.903 |
| Merlin Blankemeier et al. (2024) | 0.337 | 0.728 | 0.709 | 0.363 | 0.770 | 0.721 | 0.371 | 0.762 | 0.896 |
| T3D Liu et al. (2023) | 0.351 | 0.737 | 0.725 | 0.384 | 0.775 | 0.805 | 0.395 | 0.802 | 0.919 |
| MRM Zhou et al. (2023) | 0.276 | 0.673 | 0.652 | 0.302 | 0.735 | 0.794 | 0.324 | 0.748 | 0.885 |
| IMITATE Liu et al. (2024) | 0.295 | 0.689 | 0.664 | 0.314 | 0.724 | 0.768 | 0.330 | 0.743 | 0.897 |
| fVLM Shui et al. (2025) | 0.379 | **0.778** | **0.751** | 0.392 | 0.783 | 0.772 | 0.374 | 0.794 | 0.899 |
| SegVL (ours) | **0.382** | 0.767 | 0.740 | **0.398** | **0.793** | **0.832** | **0.413** | **0.811** | **0.924** |

Table 4: Radiology report generation comparison on the CT-RATE validation set with LLM frozen.

| Method | BLEU-1 | METEOR | ROUGE-L | CIDEr | Llama Score |
|---|---|---|---|---|---|
| Supervised | 45.8 | 27.8 | 46.7 | 2.32 | 4.87 |
| MAE He et al. (2022) | 47.6 | 30.1 | 48.1 | 2.18 | 4.96 |
| CT-CLIP Hamamci et al. (2024) | 48.8 | 30.4 | 48.3 | 2.66 | 5.44 |
| fVLM Shui et al. (2025) | 49.7 | 30.9 | 52.4 | 2.79 | 5.93 |
| **Ours** | **51.2** | **31.3** | **54.7** | **2.84** | **6.25** |

Table 5: Downstream segmentation performance under the finetuning setting.

| Method | TotalSeg | MSD-Lung |
|---|---|---|
| Supervised | 0.858 | 0.631 |
| CT-CLIP Hamamci et al. (2024) | 0.867 | 0.684 |
| MAE3D Chen et al. (2023) | 0.865 | 0.674 |
| Merlin Blankemeier et al. (2024) | 0.870 | 0.689 |
| T3D Liu et al. (2023) | \ | 0.701 |
| fVLM Shui et al. (2025) | 0.874 | 0.686 |
| UniMiSS Xie et al. (2022) | 0.868 | 0.677 |
| **SegVL (ours)** | **0.889** | **0.729** |

## 4.3 REPORT GENERATION

As shown in Table 4, our method improves upon strong baselines across both text metrics and the LLM-based clinical metric. Relative to the best prior baseline (fVLM), we observe +1.5 BLEU-1 (51.2 vs. 49.7), +0.4 METEOR (31.3 vs. 30.9), +2.3 ROUGE-L (54.7 vs. 52.4), +0.05 CIDEr (2.84 vs. 2.79), and +0.32 Llama score (6.25 vs. 5.93). These gains are achieved under a minimalist setup that *keeps the LLM frozen*, indicating that the proposed encoder learns transferable and clinically meaningful visual representations that support detailed report generation without task-specific instruction tuning.

## 4.4 RESULTS ON SEMANTIC SEGMENTATION TASKS

We evaluate segmentation transferability on two downstream benchmarks, and full-finetuning results are shown in Table 5.

**TotalSegmentator.** We use the organ subset from TotalSegmentator, which focuses on large, well-defined anatomical structures. SegVL achieves the best performance, obtaining 0.889 Dice in finetuning. It surpasses both CT-CLIP and fVLM, the latter of which explicitly aligns organ-level regions with textual descriptions. This indicates that our voxel-level contrastive supervision captures organ semantics more effectively.

**MSD-Lung.** This task involves segmenting lung tumors, which often appear as small, fine-grained lesions. SegVL achieves 0.729 Dice in finetuning, significantly outperforming fVLM and CT-CLIP by +3.6% and +4.5%, respectively. These results demonstrate the superiority of our fine-grained feature learning in detecting small pathological regions.

More results, including those under the linear probing setting, are provided in the Appendix Table 8.

## 4.5 ZERO-SHOT SEGMENTATION RESULTS

Our framework formulates segmentation as a voxel-mask contrastive task, enabling zero-shot segmentation on unseen data, with the goal of validating the alignment of fine-grained image-text features learned by our model. Given a new volume and a target anatomical label, SegVL computes the cosine similarity between each voxel embedding $\mathbf{V}_i$ and the corresponding mask embedding $\mathbf{M}_{\text{mask},j}$ as in Eq. 4, resulting in a voxel-wise probability map $P_{i,j}$. We visualize this map overlaid on the original CT slices, where high-probability regions align with expected anatomical structures. As shown in Figure 3, despite using a lightweight segmentation head, our model's predictions closely match anatomical regions, particularly for localized targets like lung nodules and effusions. Detailed quantitative results across different datasets are provided in the Appendix E.2.

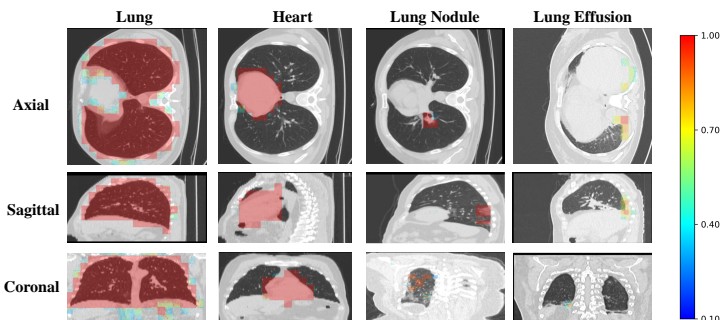

Figure 3: **Visualization of zero-shot segmentation predictions for different anatomical targets across three views.** Heatmaps indicate voxel-wise probabilities based on cosine similarity between voxel and mask embeddings, with warmer colors representing higher probabilities.

Table 6: Ablation studies of segmentation supervision, loss functions, and decoder designs on CT-RATE classification. Seg denotes whether segmentation data are incorporated during pretraining.

| Seg | VE | Loss | Decoder | Zero-shot | Lipro | Finetune |
|-----|-----|------|---------|-----------|-------|----------|
| × | × | \ | Light MLP | 0.672 | 0.683 | 0.744 |
| ✓ | × | Tversky | Light MLP | 0.707 | 0.760 | 0.795 |
| ✓ | ✓ | InfoNCE | Light MLP | 0.676 | 0.688 | 0.742 |
| ✓ | ✓ | Tversky | Heavy CNN | 0.737 | 0.752 | 0.775 |
| ✓ | ✓ | Dice | Light MLP | 0.735 | 0.763 | 0.797 |
| ✓ | ✓ | Tversky | Light MLP | **0.742** | **0.776** | **0.803** |

## 4.6 ABLATION STUDIES

**Effect of segmentation supervision.**    Comparing the baseline without segmentation (row 1) and the model trained with segmentation data (row 2), we observe substantial gains in classification. Finetuning AUC improves from 0.744 to 0.795, and linear probing also benefits. This shows that segmentation supervision enhances the encoder's *understanding ability*, guiding it toward fine-grained anatomical regions relevant for diagnosis rather than merely serving as an additional output task.

**Impact of segmentation loss.**    Among different segmentation losses, Tversky consistently yields the strongest improvements in classification (0.803 finetuned AUC), while Dice is slightly weaker and InfoNCE performs the worst. The contrastive voxel-level formulation of InfoNCE amplifies class imbalance, leading to degraded transfer performance. In contrast, Tversky handles imbalance better, thereby providing more useful supervision for representation learning.

**Effect of decoder complexity.**    Replacing the lightweight MLP decoder with a heavy CNN (row 4) results in worse classification performance. This suggests that complex decoders tend to absorb fine-grained information themselves, limiting what the encoder learns. In contrast, a lightweight decoder forces more information into the encoder, which leads to stronger and more transferable representations for downstream understanding tasks.

## 5 CONCLUSION

We present `SegVL`, a unified contrastive learning framework that incorporates segmentation supervision into vision-language pretraining for 3D medical images. By introducing voxel-level contrastive learning between voxel embeddings and segmentation prompts, and enhancing image-text contrast through segmentation-aware fusion, our model captures fine-grained anatomical features that improve performance across both classification and segmentation tasks. Extensive experiments show that `SegVL` outperforms existing 3D Med-VLP methods, especially in fine-grained recognition scenarios.

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

## A  CLARIFICATION ON RESEARCH GOALS

The primary goal of our work is to advance *understanding* in 3D medical vision–language pretraining (VLP). Our motivation is not to design a new state-of-the-art segmentation model, but to demonstrate that segmentation supervision can serve as an effective signal to enhance fine-grained understanding. Specifically, our framework unifies voxel-level contrastive learning with global image–text supervision in a mutually beneficial way: segmentation improves the model's grounding of localized semantic features, while VLP enriches the encoder's contextual representations, which in turn boost segmentation transferability.

Our experiments show state-of-the-art performance on VLP understanding benchmarks, confirming the effectiveness of our approach in capturing clinically meaningful representations. At the same time, our model also exhibits strong segmentation capability under linear probing and finetuning, out-performing prior VLP baselines. Importantly, we do not claim to surpass task-specific segmentation models such as nnU-Net Isensee et al. (2021) or specialized architectures; instead, we emphasize that segmentation in our framework is used primarily as an auxiliary signal to improve understanding, while naturally enhancing segmentation transferability within the VLP family.

## B  NETWORK IMPLEMENTATION DETAILS

**Vision Encoder**  We adopt a clean `ViT-Base` architecture as our 3D vision encoder, which directly applies standard Transformer blocks over volumetric patches. Specifically, the input CT volume is tokenized into non-overlapping $20 \times 20 \times 10$ patches, resulting in volumetric tokens. Unlike CT-ViT in Hamamci et al. (2024), we remove the preceding convolutional layers and the large MLP expansion module, preserving a pure ViT design for better modularity and generalization. The model comprises 12 Transformer layers, each with 12 attention heads and hidden size 768, matching the original ViT-Base Dosovitskiy et al. (2020) configuration. Due to the long sequence length of volumetric tokens, we adopt PyTorch's FlashAttention v2 Dao (2024) for the vision transformer to reduce memory usage without precision loss.

**Text Encoder**  The text encoder is a pre-trained transformer-based model following Hamamci et al. (2024).

**Text and Segmentation Heads.**  Both the text projection head and segmentation head used for contrastive learning are implemented as lightweight 2-layer MLPs. Each MLP comprises a linear projection, followed by a LeakyReLU activation, LayerNorm, and a final linear layer to produce the embedding. These heads are designed to be parameter-efficient and compatible with the contrastive objectives.

**Segmentation Decoder for Fine-tuning.**  When fine-tuning on downstream segmentation tasks, we replace the 2-layer segmentation MLP head with a small transposed convolutional decoder following UniMiSS Xie et al. (2022). This adjustment introduces spatial positional bias into the predictions, which is important for accurate voxel-level decoding. Without this change, we observe that shared MLP heads tend to generate overly uniform predictions across all voxels in a token due to the lack of location-specific modeling.

**Implementations of CLIP Loss on Multi-GPU**  The official CT-CLIP Hamamci et al. (2024) code does not support multi-GPU training for the CLIP loss, as it fails to perform contrastive learning across image/text embeddings from different GPUs. We address this issue by adapting the original CLIP Radford et al. (2021) implementation to HuggingFace Accelerate library Gugger et al. (2022) in all our experiments, including ablation studies. This limitation has also been resolved in the fVLM Shui et al. (2025) implementation, which is adapted from a different codebase.

## C  TRAINING IMPLEMENTATION DETAILS

**Initialization and Training Stages.**  The image encoder is randomly initialized, while the text encoder is initialized from BioViL Boecking et al. (2022). To ensure training stability and effective

representation learning, we adopt a two-stage training strategy. In early training stages, the image and text modalities are not yet semantically aligned, and the embeddings of mask label names may not carry meaningful information. Introducing segmentation supervision at this point, especially voxel-level supervision, can interfere with the learning dynamics, as segmentation is inherently a more complex and fine-grained task.

To address this, the first stage focuses solely on learning high-level semantic representations: we set $\alpha_{\text{seg}} = 0$ in Section 3.3, and keep the segmentation/text heads frozen. In the second stage, once a robust global image-text alignment has been established, we introduce segmentation supervision to refine fine-grained feature learning, setting $\alpha_{\text{seg}} = 0.75$. The fusion weight $\lambda$ (Section 3.3) is also initialized to 0 at the beginning of the second stage. This design allows the model to first acquire foundational semantic alignment, then progressively tackle the more challenging voxel-level task, leading to more stable optimization and better overall representation quality.

**Sampling Different Types of Training Dataset**   As we use both image-report and image-segmentation dataset for pre-training, the sampling and balancing strategy between two types of dataset become crucial. We perform gradient accumulation of 2 steps, which contain one step of training on image-report dataset and one step of training on image-segmentation dataset. For the image-report dataset, the segmentation head is also used for visual enhancement on image latents.

**Optimizing Hyper-parameters.**   Each stage is trained for $10^6$ steps using the AdamW optimizer and a cosine learning rate schedule, with the learning rate decaying linearly from $5 \times 10^{-6}$ to $5 \times 10^{-8}$. The batch size is set to 32 in the first stage and 16 in the second. Gradient accumulation with a step size of 2 is used in both stages to balance GPU memory constraints.

**Loss Hyper-parameters.**   For segmentation loss, we use the Tversky loss with class-specific $(\alpha, \beta)$ weights to handle severe class imbalance: $(0.3, 0.7)$ for lung nodules, $(0.4, 0.6)$ for lung effusion, and $(0.5, 0.5)$ for other categories.

**Data Augmentations.**   We follow fVLM  Shui et al. (2025) and apply standard augmentations including random flipping and random spatial shifting during the second stage. The total training time on the CT-RATE and RadGenome datasets is approximately 4 days for each stage using 4 NVIDIA A100 GPUs (80GB) with mixed-precision training enabled.

# D  DATA SETTINGS

## D.1  PRE-TRAINING DATASET SETTINGS

The segmentation classes used for pre-training in RadGenome-ChestCT dataset include lung, heart, tracheobronchial tree, cardiovascular, lung nodule and lung effusion.

Due to the low-quality of lung nodule masks in RadGenome-ChestCT dataset, these masks are refined using trained nnU-Net  Isensee et al. (2021) model on LUNA  Setio et al. (2017) dataset.

Additionally, due to the high false positive rate in lung effusion labels, we discard all corresponding label masks for CT volumes with negative label of "no pleural effusion," replacing them with all-background masks.

The tracheobronchial tree mask is constructed by merging the "trachea" and "bronchi" masks. The cardiovascular mask is created by combining anatomical structures including the aorta, aortic arch, brachiocephalic trunk, brachiocephalic vein, carotid artery, common carotid artery, heart ascending aorta, heart atrium, heart ventricle, heart, inferior vena cava, internal carotid artery, and internal jugular vein.

## D.2  DOWNSTREAM DATASET AND EVALUATION TASKS

**CT-RATE Hamamci et al. (2024) (Internal Classification)**   The CT-RATE internal validation dataset consists of 3039 volumes from 1304 patients. The evaluated classes for classification follows Hamamci et al. (2024), including 18 classes of Medical material, Arterial wall calcification, Cardiomegaly, Pericardial effusion, Coronary artery wall calcification, Hiatal hernia, Lymphadenopathy,

Emphysema, Atelectasis, Lung nodule, Lung opacity, Pulmonary fibrotic sequela, Pleural effusion, Mosaic attenuation pattern, Peribronchial thickening, Consolidation, Bronchiectasis, Interlobular septal thickening.

**Rad-ChestCT Draelos et al. (2021) (External Classification)**    The Rad-ChestCT dataset consists of 3630 CT volumes. The evaluated classes for classification follows Hamamci et al. (2024), including 16 classes of Medical material, Calcification, Cardiomegaly, Pericardial effusion, Hiatal hernia, Lymphadenopathy, Emphysema, Atelectasis, Lung nodule, Lung opacity, Pulmonary fibrotic sequela, Pleural effusion, Peribronchial thickening, Consolidation, Bronchiectasis and Interlobular septal thickening.

**CC-CCII Zhang et al. (2020) (External Classification)**    The CC-CCII dataset consists of 3,993 scans from 2,698 patients, and we perform classification following the settings of He et al. (2024). The downstream task is to classify each volume into three categories: novel coronavirus pneumonia (NCP), common pneumonia (CP), and normal (Normal).

**RICORD Tsai et al. (2020) (External Classification)**    The RICORD dataset comprises 182 training volumes and 45 testing volumes. Following the protocol of UnimiSS Xie et al. (2022), we formulate the classification task as a binary prediction of COVID-19 positivity.

**MSD-Lung Antonelli et al. (2022) (External Segmentation)**    The Medical Segmentation Decathlon (MSD) challenge dataset comprises CT volumes with lesion and its segmentation. We use the task 6 split of MSD challenge dataset to focus on the lung tumor segmentation on chest CT. The dataset comprises 63 training volumes with lung tumor annotations and 32 testing volumes. We follow the evaluation settings in VoCo Wu et al. (2024).

**TotalSegmentator Wasserthal et al. (2023) (External Segmentation)**    We use the organ subset of the TotalSegmentator dataset for evaluation. The organ subset includes segmentations of Spleen, Left & Right Kidney, Gallbladder, Liver, Stomach, Pancreas, Left & Right Adrenal Gland, Lobes of Left & Right Lung, Esophagus, Trachea, Thyroid Gland, Small Bowel, Duodenum, Colon, Urinary Bladder, Prostate, Left & Right Kidney Cyst. We split the dataset into 928 training, 52 validation, and 248 testing volumes.

**RadGenome Zhang et al. (2024) (Zero-Shot Segmentation)**    We perform zero-shot segmentation on the RadGenome validation set for qualitative visualization. The evaluated classes include lung, heart, lung nodule, and lung effusion. Heatmaps are generated to visualize voxel-level segmentation predictions for each class.

# E    MORE ZERO-SHOT SEGMENTATIONS RESULTS

## E.1    VISUALIZATIONS OF ZERO-SHOT SEGMENTATIONS

To supplement our analysis of voxel-mask contrastive learning, we include additional zero-shot segmentation results in Figure 4–6. The visualizations display predicted probability maps obtained from cosine similarity between voxel and mask embeddings. These examples further confirm that our method produces semantically meaningful and spatially precise segmentations, even for subtle or fine-grained structures, without relying on heavy decoder designs.

## E.2    QUANTITATIVE EVALUATION OF ZERO-SHOT SEGMENTATION

We additionally provide quantitative results here. We emphasize that these experiments are intended to validate semantic alignment in the learned representations rather than to claim state-of-the-art zero-shot segmentation accuracy.

Following prior works Howlader et al. (2024); Stojnić et al. (2025), we adopt *patch-level* Dice and IoU as evaluation metrics, which are suitable for representation-level assessment without convolutional decoders. Since our pretraining employs only lightweight MLP heads, the outputs remain at the

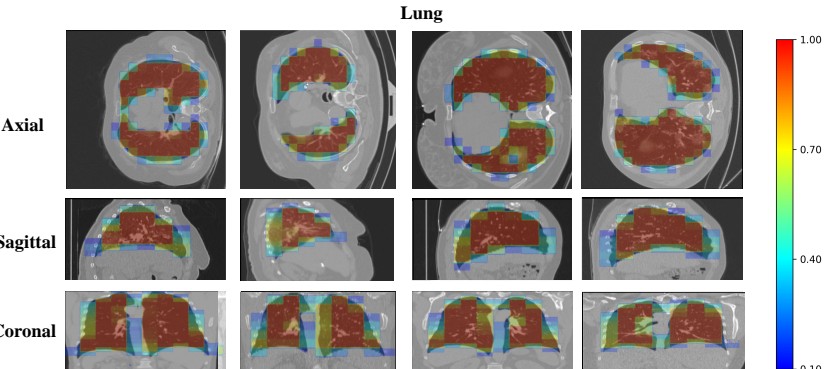

Figure 4: **Additional visualization of lung predictions.** Segmentation heatmaps predicted by SegVL for the *lung* class across axial, sagittal, and coronal views. Warmer colors indicate higher predicted probabilities, showing close alignment with lung regions.

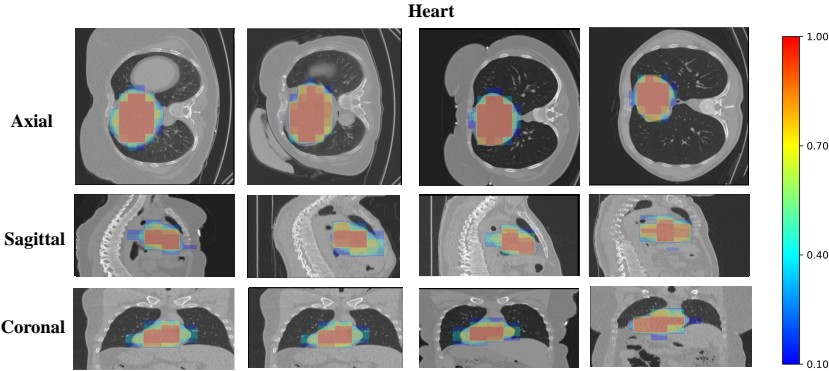

Figure 5: **Additional visualization of heart predictions.** Segmentation heatmaps predicted by SegVL for the *heart* class across axial, sagittal, and coronal views. Our model accurately highlights heart regions using only lightweight contrastive supervision.

patch/token level, unlike linear-probing or finetuning settings. This design emphasizes semantic understanding rather than task-specific decoder performance.

We report results on two public datasets included in our study, restricted to anatomical classes overlapping with our training set: Lung and Heart on TotalSegmentator, and Lung Nodule on MSD-Lung. Results are summarized in Table 7. Our approach achieves strong patch-level alignment on both datasets, with Dice of 0.904 (TotalSegmentator) and 0.776 (MSD-Lung). These results indicate that the learned representations capture discriminative, clinically relevant features that generalize to unseen test data.

Table 7: Patch-level zero-shot segmentation results. We report Dice and IoU on overlapping classes of two datasets. Metrics reflect representation quality rather than full voxel-wise segmentation.

| Dataset | Dice | IoU |
|---|---|---|
| TotalSegmentator (Lung, Heart) | 0.904 | 0.831 |
| MSD-Lung (Lung Nodule) | 0.776 | 0.645 |

## F  LINEAR-PROBING SEGMENTATION RESULTS

Following UniMiSS Xie et al. (2022), we adopt a lightweight convolution-layer based segmentation head for the linear probing setting.

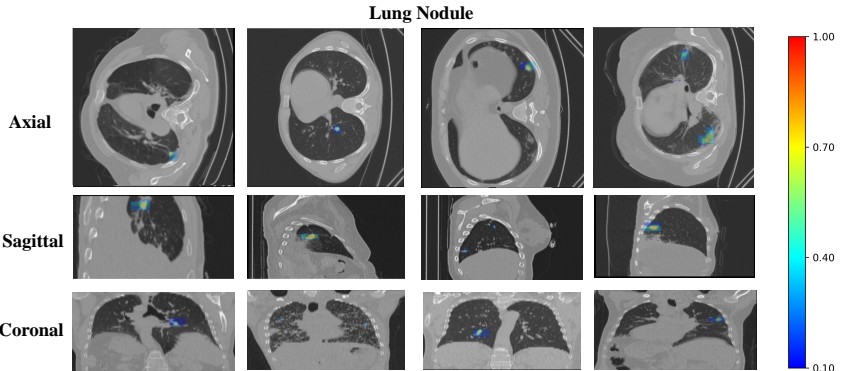

Figure 6: **Additional visualization of lung nodule predictions.** Visualization of voxel-wise prediction maps for the *lung nodule* class. The highlighted small regions reflect our model's ability to localize subtle, fine-grained features via voxel-mask contrastive learning.

Table 8: Downstream segmentation performance under the linear probing setting.

| Method | TotalSeg | MSD-Lung |
|--------|----------|----------|
| CT-CLIPHamamci et al. (2024) | 0.852 | 0.637 |
| MerlinBlankemeier et al. (2024) | 0.859 | 0.645 |
| fVLMShui et al. (2025) | 0.861 | 0.639 |
| UniMiSSXie et al. (2022) | 0.854 | 0.642 |
| SegVL (**ours**) | **0.878** | **0.675** |

# G    ADDITIONAL RESULTS ON THE AMOS DATASET

To validate the transferability to abdominal CT, we further compare our methods with SOTA VLPs trained on chest-CT data. As shown in Table 9, our method achieves competitive performance on AMOS. Compared to prior VLP baselines (CTCLIPHamamci et al. (2024) and fVLMShui et al. (2025)), SegVL provides a clear improvement (+1.5 Mean Dice over CTCLIP, +0.9 over fVLM). Although nnU-Net Isensee et al. (2021) trained from scratch on AMOS achieves the highest absolute score (0.908), it requires task-specific training, while our approach transfers directly from CT-RATE pretraining.

| Method | Settings | Mean Dice ↑ |
|--------|----------|-------------|
| CTCLIP | VLP, Pretrained on CT-RATE (lung) | 0.858 |
| fVLM | VLP, Pretrained on CT-RATE (lung) | 0.866 |
| nnU-Net | nnU-Net, Trained from scratch (Abdomen) | 0.908 |
| **SegVL (Ours)** | VLP, Pretrained on CT-RATE (lung) | **0.881** |

Table 9: Additional results on the AMOS dataset for abdominal organ segmentation. SegVL achieves consistent gains over prior VLP baselines, demonstrating its ability to generalize across domains.

# H    PERFORMANCE COMPARISON WITH STANDARD DEVIATION

We provide the performance comparisons with standard deviation in our main experiments of downstream tasks evaluation. The deviation is obtained from five repeated experiments with different random seeds.

Table 10: Mean AUC (↑) and estimated standard deviation on downstream classification under **linear probing** setting.

| Model | Rad-ChestCT | CC-CCII | RICORD |
|---|---|---|---|
| CT-CLIP Hamamci et al. (2024) | $0.653 \pm 0.014$ | $0.865 \pm 0.009$ | $0.846 \pm 0.008$ |
| Merlin Blankemeier et al. (2024) | $0.677 \pm 0.011$ | $0.877 \pm 0.008$ | $0.854 \pm 0.007$ |
| UniMiSS Xie et al. (2022) | \ | $0.841 \pm 0.009$ | $0.862 \pm 0.008$ |
| fVLM Shui et al. (2025) | $0.697 \pm 0.010$ | $0.871 \pm 0.007$ | $0.858 \pm 0.007$ |
| SegVL (ours) | $\mathbf{0.702 \pm 0.009}$ | $\mathbf{0.893 \pm 0.006}$ | $\mathbf{0.887 \pm 0.007}$ |

Table 11: Comparison of our SegVL with other baselines on downstream classification tasks under **finetuning** setting. Results are reported as AUC (↑) in the format of mean ± std. Values marked with n/a are directly taken from original papers without standard deviation.

| Model | Rad-ChestCT | CC-CCII | RICORD |
|---|---|---|---|
| CT-CLIP Hamamci et al. (2024) | $0.650 \pm 0.011$ | $0.920 \pm 0.007$ | $0.863 \pm 0.006$ |
| Merlin Blankemeier et al. (2024) | $0.694 \pm 0.012$ | $0.919 \pm 0.009$ | $0.879 \pm 0.010$ |
| T3D Liu et al. (2023) | \ | $0.927 \pm$ n/a | \ |
| UniMiSS Xie et al. (2022) | \ | $0.913 \pm 0.008$ | $0.891 \pm 0.009$ |
| MRM Zhou et al. (2023) | \ | $0.880 \pm$ n/a | \ |
| IMITATE Liu et al. (2024) | \ | $0.892 \pm$ n/a | \ |
| fVLM Shui et al. (2025) | $0.704 \pm 0.010$ | $0.926 \pm 0.007$ | $0.885 \pm 0.006$ |
| SegVL (ours) | $\mathbf{0.716 \pm 0.008}$ | $\mathbf{0.936 \pm 0.006}$ | $\mathbf{0.912 \pm 0.005}$ |

## I    DETAILED ANALYSIS OF CLASSIFICATION RESULTS OF DIFFERENT LABELS

We present detailed zero-shot and linear probing classification results on the CT-RATE dataset in Table 12 and Table 13 Overall, our model achieves strong performance across 18 diagnostic categories, demonstrating the generalizability of our pretraining approach.

Two key observations emerge from this analysis. First, the incorporation of segmentation supervision improves the model's understanding of anatomy-related diseases. For instance, we observe notable gains on classes such as *bronchiectasis* and *lung nodule*, which are highly correlated with anatomical structures included in our segmentation vocabulary.

Second, our method exhibits clear advantages in identifying fine-grained patterns. Categories like *lung opacity*, which require subtle feature discrimination, benefit from the detailed spatial information learned through voxel-level contrastive supervision. This suggests that our approach not only enhances high-level semantic alignment but also reinforces the model's sensitivity to nuanced radiological cues.

## J    LIMITATIONS AND FUTURE WORK.

Our current study employs a larger set of segmentation classes than prior organ-level VLP work. For instance, fVLM Shui et al. (2025) only utilized four organ categories on CT-RATE for organ-level contrastive learning, whereas we include additional classes to more comprehensively assess the effect of segmentation supervision on representation learning. At this stage, we deliberately restrict the class set to those with reliable, radiologist-confirmed annotations, so as to provide a clean validation of whether segmentation signals can enhance understanding without being confounded by noisy labels. In future work, we plan to broaden the anatomical vocabulary and investigate the use of semi-supervised techniques (e.g., consistency-based objectives) and robust training strategies for noisy labels, to better leverage large-scale but imperfect segmentation resources under constrained annotation settings.

## K    POTENTIAL SOCIETAL IMPACT.

Our work presents a pre-trained 3D medical image encoder that may benefit a wide range of downstream applications, such as computer-aided diagnosis and clinical decision support. By enabling better understanding of volumetric scans with limited annotations, our approach could help democratize access to high-quality medical AI systems and reduce the burden on radiologists.

Table 12: Detailed zero-shot classification results on CT-RATE. We report class-wise Precision, AUC, and F1 score for each of the 18 diagnostic categories, along with their mean.

| Class | Precision | AUC | F1 |
|---|---|---|---|
| Medical material | 0.247 | 0.742 | 0.700 |
| Arterial wall calcification | 0.581 | 0.890 | 0.831 |
| Cardiomegaly | 0.557 | 0.880 | 0.911 |
| Pericardial effusion | 0.477 | 0.891 | 0.849 |
| Coronary artery wall calcification | 0.548 | 0.839 | 0.779 |
| Hiatal hernia | 0.185 | 0.738 | 0.655 |
| Lymphadenopathy | 0.416 | 0.675 | 0.700 |
| Emphysema | 0.358 | 0.759 | 0.743 |
| Atelectasis | 0.372 | 0.691 | 0.705 |
| Lung nodule | 0.581 | 0.702 | 0.658 |
| Lung opacity | 0.522 | 0.711 | 0.659 |
| Pulmonary fibrotic sequela | 0.359 | 0.590 | 0.554 |
| Pleural effusion | 0.480 | 0.931 | 0.918 |
| Mosaic attenuation pattern | 0.132 | 0.714 | 0.671 |
| Peribronchial thickening | 0.249 | 0.719 | 0.698 |
| Consolidation | 0.356 | 0.809 | 0.759 |
| Bronchiectasis | 0.240 | 0.775 | 0.751 |
| Interlobular septal thickening | 0.218 | 0.756 | 0.786 |
| Mean | 0.382 | 0.767 | 0.740 |

Table 13: Per-class linear probing results on the CT-RATE dataset. We report class-wise Precision, AUC, and F1 score for each of the 18 diagnostic categories, along with their mean.

| Class | Precision | AUC | F1 |
|---|---|---|---|
| Medical material | 0.286 | 0.780 | 0.809 |
| Arterial wall calcification | 0.575 | 0.853 | 0.847 |
| Cardiomegaly | 0.387 | 0.915 | 0.924 |
| Pericardial effusion | 0.420 | 0.912 | 0.958 |
| Coronary artery wall calcification | 0.550 | 0.869 | 0.876 |
| Hiatal hernia | 0.257 | 0.759 | 0.865 |
| Lymphadenopathy | 0.456 | 0.799 | 0.810 |
| Emphysema | 0.310 | 0.720 | 0.828 |
| Atelectasis | 0.480 | 0.815 | 0.844 |
| Lung nodule | 0.636 | 0.725 | 0.791 |
| Lung opacity | 0.520 | 0.749 | 0.781 |
| Pulmonary fibrotic sequela | 0.385 | 0.720 | 0.739 |
| Pleural effusion | 0.680 | 0.929 | 0.943 |
| Mosaic attenuation pattern | 0.227 | 0.754 | 0.808 |
| Peribronchial thickening | 0.184 | 0.699 | 0.762 |
| Consolidation | 0.374 | 0.839 | 0.876 |
| Bronchiectasis | 0.216 | 0.641 | 0.697 |
| Interlobular septal thickening | 0.217 | 0.791 | 0.812 |
| Mean | 0.398 | 0.793 | 0.832 |

However, potential negative impacts should be considered. The pre-trained model may reflect dataset biases, such as under-representation of rare conditions or specific demographic groups, which could lead to reduced accuracy or unintended disparities in clinical settings. Moreover, over-reliance on automated systems without sufficient human oversight may risk diagnostic errors. Careful evaluation and responsible deployment in real-world workflows are necessary to mitigate such risks.

## L LICENSES

We used existing assets as follows:

- **CT-CLIP** Hamamci et al. (2024): `https://github.com/ibrahimethemhamamci/CT-CLIP`, licensed under CC BY-NC-SA.
- **fVLM** Shui et al. (2025): `https://github.com/alibaba-damo-academy/fvlm`, no explicit license.
- **UniMiSS** Xie et al. (2022): `https://github.com/YtongXie/UniMiSS-code`, licensed under MIT.
- **Merlin** Blankemeier et al. (2024): `https://github.com/StanfordMIMI/Merlin`, licensed under MIT.

