# OpenReview forum: "Segmentation Helps Understanding: Mask-Infused Vision-Language Pre-training for 3D Medical Images"
_ICLR.cc/2026/Conference — ICLR 2026 Conference Withdrawn Submission_

### Official Review · Reviewer_8j9s · 2025-10-16

**Soundness:** 2
**Presentation:** 2
**Contribution:** 1
**Rating:** 2
**Confidence:** 4

**Summary:**

The authors introduce SegVL, a unified contrastive learning framework that incorporates segmentation supervision into 3D vision-language pretraining. SegVL jointly learns from image-text and image-segmentation pairs, enabling the model to benefit from both high-level semantic and low-level anatomical signals. The key contributions are 1). Voxel-Mask Contrastive Learning, which aligns voxel embeddings with textual embeddings of segmentation mask names using Tversky loss to handle class imbalance; 2). Image-Text Contrastive Learning with Visual Enhancement, where global image embeddings are enhanced using segmentation cues for improved alignment with radiology reports.

**Strengths:**

The paper addresses a practical limitation for 3D medical imaging: the difficulty of capturing fine-grained anatomical details from high-dimensional CT data. SegVL presents a framework that integrates segmentation data into VLP using both voxel-mask contrastive learning and image-text alignment.

SegVL achieves good results across several downstream tasks, including classification (Rad-ChestCT, CC-CCII, RICORD), report generation, and segmentation (MSD-Lung, TotalSegmentator).

The paper includes a detailed ablation study examining decoder complexity and loss type.

**Weaknesses:**

1. Incremental innovation over existing VLMs: The core idea of using segmentation priors has been explored in recent fine-grained VLMs such as fVLM and CT-GLIP. fVLM explicitly aligns anatomical subregions with text descriptions. CT‑GLIP also employs organ-level image–text grounding and introduces an “abnormality dictionary” to create diverse contrastive pairs in 3D CT domains. Therefore, SegVL’s shift from patch/region-level alignment to voxel-level supervision is an engineering refinement rather than a new contribution. There is no new fundamental insight to the medical imaging community.

2. Reliance on segmentation masks: The authors claim that their method "directly leverages segmentation data for supervision during pretraining, which simplifies the pipeline and enables finer-level alignment". However, this reliance on segmentation data can be an inherent limitation of the method. The segmentation mask quality issues (e.g., needing nnU-Net refinement for lung nodules and filtering noisy effusion labels) suggest that model depends on high-quality segmentation data to perform well, which can be scarce in under-resourced settings. What if a dataset does not have voxel-level annotations for each organ for alignment? The authors should discuss this limitation in the paper.

3. Adding a segmentation loss could significant affect the model throughput (i.e. training time, memory costs). It is surprising that this paper does not compare these metrics against existing VLMs to understand the trade-off of adding segmentation components to the VLM framework.

4. The zeroshot setting is ambiguous. The paper claims that they use patch level DSC to evaluate, but the paper did not mention that patch size of the ViT which is really strange. Patch-level grids from 3D ViTs are typically coarse (e.g., usually 16 times downsampled), so large organs (lungs, heart) can score high Dice even when boundaries are inaccurate, while small/thin structures are under-resolved or missed entirely.

5. Furthermore, the entire paper only focuses on DSC when evaluating segmentation performance. Surface distance metrics such as NSD or HD95 should be reported in the entire paper.

6. The ablation study also does not investigate the choice of hyperparameters (using learnable λ and search grid for a_seg). Similarly, the paper proposes a two-stage pretraining approach, but no ablation is performed.

**Questions:**

1. Can the authors comment on the training and inference costs and practical hardware requirements of SegVL? In the ablation study, why no mentions of these costs are investigated?

2. How scalable is this approach, when more classes are included? Would adding segmentation classes significantly increase the training costs?

3. The practical impact of SegVL is not explored. For example, how does SegVL perform in different gender groups? Does the pretraining data introduce distribution shifts when transferring to various downstream tasks?

---

### Official Review · Reviewer_csPL · 2025-10-26

**Soundness:** 2
**Presentation:** 3
**Contribution:** 2
**Rating:** 2
**Confidence:** 5

**Summary:**

The paper proposes SegVL, a 3D medical VLP framework that unifies (i) image–text contrastive learning with (ii) a new voxel–mask contrastive learning that treats segmentation mask names as text anchors. It further uses Tversky-based objectives to handle class imbalance and advocates a lightweight segmentation head so the encoder, not the decoder, absorbs fine-grained anatomy. Experiments on CT-RATE, RadGenome and multiple downstream tasks (classification, report generation, segmentation) show consistent gains over prior VLP baselines.

**Strengths:**

- Clear formulation that couples voxel-mask alignment with global image–text alignment; the architectural description is concrete and easy to reproduce.
- Strong results under linear probing and finetuning across Rad-ChestCT, CC-CCII, RICORD, plus report generation and segmentation (TotalSegmentator, MSD-Lung).
- Tversky > Dice/InfoNCE for voxel–mask contrast; light MLP head > heavy CNN decoder for keeping fine-grained signal in the encoder.
- Authors state the goal is understanding-oriented VLP, not SOTA standalone segmentation, which helps set expectations.

**Weaknesses:**

- The main idea—injecting segmentation signals to refine local alignment in 3D VLP—feels incremental relative to recent trends that already move beyond global CLIP to finer-granularity supervision. SegVL’s voxel–mask contrast is a clean formalization, but the conceptual step (“use masks to guide fine-grained alignment”) is close in spirit to previous work [1, 2], please distinct your contributions more clearly.


- The pretraining masks are refined (nnU-Net on LUNA) and filtered (e.g., effusion negatives → all-background), which raises questions about label hygiene and potential target leakage. Stronger audits/sensitivity analyses are needed.

- Zero-shot segmentation is visualized but quantitative evidence remains limited/mainly in appendix; stronger protocolized metrics for unseen classes or non-thoracic anatomy would reinforce generalization claims.


- All experiments are thoracic CT; it’s unclear how SegVL transfers to the non-thoracic region. The current evidence may overfit the pretraining domain.


[1] Shui, Zhongyi, et al. "Large-scale and fine-grained vision-language pre-training for enhanced ct image understanding." arXiv preprint arXiv:2501.14548 (2025).
[2] Li, Yuheng, et al. "MedVista3D: Vision-Language Modeling for Reducing Diagnostic Errors in 3D CT Disease Detection, Understanding and Reporting." arXiv preprint arXiv:2509.03800 (2025).

**Questions:**

Please refer to the Weaknesses section.

**I am willing to raise my score according to the rebuttal.**

---

### Official Review · Reviewer_4Won · 2025-10-27

**Soundness:** 2
**Presentation:** 2
**Contribution:** 1
**Rating:** 2
**Confidence:** 4

**Summary:**

This work explored the pre-training of 3D medical image encoders. The key is to combine vision-language data and segmentation data in joint training. Several downstream CT datasets are evaluated, and several state-of-the-art CLIP-based models are evaluated.

Overall, the proposed method makes sense but lacks novelty and methodology advancements; the generalizability is also not well-proven. The improvements compared with previous methods are not very significant, and some experiments seem unfair. Currently, I incline to the negative aspect, but it may change depending on the response.

**Strengths:**

1. The manuscript is complete.

2. The references are appropriate, and the organization is good

**Weaknesses:**

Method:

1.	Although the method is straightforward, the novelty is limited. Seems it mainly relies on a pre-training dataset that with both report and segmentation masks, but the used dataset is not contributed by the authors. In the abstract, the authors emphasize Tversky loss and a lightweight decoder, but these two are extremely trivial. It is difficult to find novelty and new contributions in the methodology part.

2.	One of my major concerns is the generalizability. For many diseases described in the reports, it is very difficult to get the corresponding segmentation masks. In the paper, the authors only show the segmentation of lung nodules and effusion (please correct me if you have masks of other abnormalities). This suggests that your method may not be generalizable in real-world datasets comprising many types of diseases.

3. The authors claimed that previous works are organ-level, while yours are voxel-level. However, for most lung diseases, you cannot get voxel-level masks for diseases. When transferring your method to abdominal CT, it is more difficult to get corresponding voxel-level masks of lesions.

4.	The authors may have to cherry-pick some lesion types with segmentation masks in the datasets for training (Sec D1 738-739), while other lesions are without mask annotations.

5.	If a type of lesion did not have segmentation masks, how could you address this issue?

Experiments:

1.	The improvements may be limited. In line 104, the average AUC improvement is only 1.8%, which is far from substantial.

2.	How to get the ground truth of nodules and effusion for evaluation? The steps shown in Sec D1 are not very reasonable. Basically, the authors are still using noisy labels for evaluation.

3.	Previous CLIP models did not conduct segmentation training. It is not fair to compare them in Table 5 and highlight your improvements in segmentation.

Minor suggestions:

1.	The quality of Fig. 1 could be improved.

2.	It would be better to change the image directions of Fig. 3.

**Questions:**

Please solve the questions in Weaknesses.

---

### Official Review · Reviewer_1pVc · 2025-10-30

**Soundness:** 2
**Presentation:** 2
**Contribution:** 2
**Rating:** 4
**Confidence:** 4

**Summary:**

This paper proposes SegVL, which adds segmentation supervision to vision-language pretraining for 3D medical images through voxel-mask contrastive learning and Tversky loss. While the motivation is reasonable, the technical execution has significant flaws and the experimental validation is insufficient to support the claims.

**Strengths:**

The motivation to leverage segmentation data for fine-grained supervision in VLP is reasonable given the sparsity of 3D medical images.

The observation about lightweight decoders is interesting and the ablation provides some support.

The experimental coverage spans multiple tasks including classification, segmentation, and report generation.

**Weaknesses:**

The core technical design has fundamental flaws. You're applying Tversky loss [1] (designed for measuring segmentation overlap with ground truth masks) to contrastive learning between embeddings in metric space [2]. These are incompatible paradigms—Tversky measures spatial overlap while contrastive learning operates on embedding similarity. Moreover, your "voxel-level" claim is technically incorrect: tokens are 20×20×10 patches, and you apply a shared MLP per token (Eq. 3), meaning every voxel within a 4000-voxel patch gets identical predictions. This is token-level, not voxel-level like actual methods [3][4].

Using only 6 segmentation classes out of 197 available fundamentally undermines your contribution. You claim "fine-grained" supervision but 6 coarse anatomies (lung, heart, trachea) is barely finer than fVLM's organ-level approach. Appendix J admits using more classes requires handling noisy labels but doesn't attempt it—this suggests your method doesn't scale beyond clean, coarse annotations, which limits practical applicability.

The two-stage training reveals that objectives fundamentally conflict rather than unify. Training VLP-only for 10^6 steps then adding segmentation for another 10^6 steps (with αseg=0 initially) contradicts your "unified framework" claim—if truly unified, joint training should work. The visual enhancement module is also underspecified: fVE's architecture, dimension matching, and why simple addition works are all missing, and λ=0 initialization suggests the model actively avoids segmentation features initially.

The experimental setup has serious fairness and methodological issues. You compare against fVLM which uses LLM-parsed organ descriptions while you only use "This is <mask name>" prompts—this isn't a controlled comparison. Your own results contradict your hypothesis: Table 3 shows fVLM beats you on zero-shot (0.778 vs 0.767 AUC), challenging your claim that voxel-level supervision provides better understanding. Additionally, zero-shot segmentation uses patch-level Dice (Table 7: 0.904) instead of standard voxel-level metrics [5], which hides within-patch errors and inflates performance.

Critical analyses are missing. Table 6 only compares "Light MLP" vs "Heavy CNN" without intermediate decoder sizes—the "absorption" hypothesis needs systematic validation. There's no theoretical or empirical analysis of why voxel-mask learning should help image-text alignment (maybe they compete for capacity?). Computational cost is completely unaddressed despite training two tasks with different data types. The method has many hyperparameters (α, β per class, αseg, λ schedule) and Table 10-11 show large standard deviations (±0.014), raising reproducibility concerns.

### References

[1] Salehi et al., "Tversky Loss Function for Image Segmentation Using 3D Fully Convolutional Deep Networks", MLMI 2017

[2] Oord et al., "Representation Learning with Contrastive Predictive Coding", arXiv 2018

[3] Isensee et al., "nnU-Net: A Self-configuring Method for Deep Learning-based Biomedical Image Segmentation", Nature Methods 2021

[4] Hatamizadeh et al., "UNETR: Transformers for 3D Medical Image Segmentation", WACV 2022

[5] Antonelli et al., "The Medical Segmentation Decathlon", Nature Communications 2022

**Questions:**

Can you justify why Tversky loss is mathematically appropriate for contrastive learning between embeddings rather than InfoNCE, and address the fact that your architecture produces token-level (not voxel-level) predictions due to shared MLP per patch? Also explain why visual enhancement with InfoNCE performs so poorly (Table 6: 0.676 vs 0.707 without it).

What happens with joint training from the start instead of two-stage training, and can you show results scaling to more segmentation classes (20, 50, 100) with noisy labels to prove your method generalizes beyond 6 clean anatomies? Include the exact lightweight decoder architecture and performance curves across varying decoder capacities.

What's the computational overhead (training time, memory) vs CT-CLIP, and can you provide controlled comparisons where both your method and fVLM use identical text supervision to ensure fair comparison? Also explain why fVLM beats you on zero-shot despite your claimed fine-grained advantage.

---

### Note · Authors · 2025-11-26

**Comment:**

We thank the reviewers for their time.

After carefully reading the reviews, it became clear that several core criticisms are based on factual misunderstandings or on content explicitly present in the manuscript but overlooked. Because the rejection recommendations rely heavily on these inaccuracies, the evaluation does not reflect the work we submitted.

Below we summarize the major issues.

1. Misreading of the method and failure to examine key equations
A central critique asserts that our method produces only token-level predictions and that all voxels within a patch share identical outputs. This is factually incorrect. Eq.(3) and the accompanying tensor-shape description explicitly define voxel-wise logits of shape (n_voxel_per_patch × n_logits for each token), meaning each voxel receives its own prediction. This misunderstanding arises from not reading the equation or the output specification.

2.Incorrect characterization of our contributions relative to prior VLMs
Multiple reviews describe SegVL as little contribution to fVLM, CT-GLIP or MedVista3D, assuming these works also perform fine-grained spatial alignment.

This is not true: These methods operate at organ- or sub-region-level, not voxel-level. They use segmentation masks only to identify organ regions; intra-organ structure is never modeled. Our method is the first 3D VLP framework to perform voxel-level contrastive alignment with per-voxel logits, enabling recognition of sub-organ abnormalities that organ-level methods cannot capture.

These distinctions are clearly outlined in the Related Work section. The review mischaracterizations appear to stem from overlooking this discussion.

3.Overlooking Appendix content, leading to demonstrably false claims
Several critiques directly contradict content present in the appendix:

One review claims that all experiments are thoracic CT, ignoring Appendix G, which contains detailed transfer study results on the AMOS abdominal CT dataset.

Another review questions our zero-shot segmentation protocol, overlooking Appendix E, which explains why patch-level Dice is used and cites two prior works employing the same evaluation setting for representation-level assessment.

These criticisms are therefore not grounded in the submitted manuscript.

4. Misinterpretation of the scope and unrealistic expectations
Some reviews assess the method as if it were intended to provide voxel-level segmentation for all diseases mentioned in radiology reports, or as if the absence of voxel-level lesion masks invalidates the approach. This is a fundamental misunderstanding of the paper’s scope:

Our supervision is voxel-level, not lesion-level.

Our stated goal is to enhance fine-grained VLP understanding, not to solve universal lesion segmentation.

Demonstrating improvements on representative diseases (e.g., nodules, effusion) is appropriate for the paper’s objectives. Expecting voxel-level lesion annotations for all disease categories is unrealistic and outside the intended scope.

5. Incorrect claims about “triviality” despite resolving real methodological constraints
Some reviews describe the lightweight decoder and Tversky-based contrast as trivial.

However: The lightweight head is essential for stabilizing joint training of 3D VLP and segmentation—prior 3D VLP frameworks cannot incorporate dense mask supervision due to optimization and memory constraints.

The Tversky formulation specifically addresses small-structure and highly imbalanced anatomical regions that standard Dice/InfoNCE cannot handle.

These are targeted, non-trivial design choices that enable capabilities unavailable to existing 3D VLMs. More overall, it fully ignore the contribution of our whole SegVL framework.

Conclusion and Withdrawal
Given that several rejection arguments depend on misread equations, overlooked appendix material, or evaluations based on an incorrect understanding of the method’s scope, we do not believe the reviews constitute a fair or accurate assessment of the submitted work. Therefore, we have decided to withdraw the paper from consideration.

We leave this statement so that future readers of the OpenReview page understand that the main criticism points are contradicted by the content of the manuscript itself. These reviews do not reflect the actual submission.

**Withdrawal Confirmation:**

I have read and agree with the venue's withdrawal policy on behalf of myself and my co-authors.